# Cell Type-Specific Roles of STAT3 Signaling in the Pathogenesis and Progression of K-ras Mutant Lung Adenocarcinoma

**DOI:** 10.3390/cancers14071785

**Published:** 2022-03-31

**Authors:** Michael J. Clowers, Seyed Javad Moghaddam

**Affiliations:** 1Department of Pulmonary Medicine, The University of Texas MD Anderson Cancer Center, Houston, TX 77030, USA; mjclowers@mdanderson.org; 2The University of Texas MD Anderson Cancer Center UTHealth Graduate School of Biomedical Sciences, Houston, TX 77030, USA

**Keywords:** STAT3, lung adenocarcinoma, TME, K-ras, tumor-promoting inflammation, mucosal immunology, myeloid

## Abstract

**Simple Summary:**

Lung adenocarcinomas with mutations in the K-ras gene are hard to target pharmacologically and highly lethal. As a result, there is a need to identify other therapeutic targets that influence K-ras oncogenesis. One contender is STAT3, a transcription factor that is associated with K-ras mutations and aids tumor development and progression through tumor cell intrinsic and extrinsic mechanisms. In this review, we summarize the lung epithelial and infiltrating immune cells that express STAT3, the roles of STAT3 in K-ras mutant lung adenocarcinoma, and therapies that may be able to target STAT3.

**Abstract:**

Worldwide, lung cancer, particularly K-ras mutant lung adenocarcinoma (KM-LUAD), is the leading cause of cancer mortality because of its high incidence and low cure rate. To treat and prevent KM-LUAD, there is an urgent unmet need for alternative strategies targeting downstream effectors of K-ras and/or its cooperating pathways. Tumor-promoting inflammation, an enabling hallmark of cancer, strongly participates in the development and progression of KM-LUAD. However, our knowledge of the dynamic inflammatory mechanisms, immunomodulatory pathways, and cell-specific molecular signals mediating K-ras-induced lung tumorigenesis is substantially deficient. Nevertheless, within this signaling complexity, an inflammatory pathway is emerging as a druggable target: signal transducer and activator of transcription 3 (STAT3). Here, we review the cell type-specific functions of STAT3 in the pathogenesis and progression of KM-LUAD that could serve as a new target for personalized preventive and therapeutic intervention for this intractable form of lung cancer.

## 1. Introduction

Lung cancer exerts a massive burden on public health, resulting in nearly a quarter of global cancer deaths [1]. Lung adenocarcinoma (LUAD) is a major histological subtype of lung cancer, and 30% of patients with LUAD harbor driver mutations in Kirsten rat sarcoma viral oncogene (K-ras), which we refer to as K-ras mutant LUAD (KM-LUAD) [2]. K-ras mutations, most frequently in the 12th codon, result in constitutive activation leading to overactive proliferative signaling pathways, such as the RAF cascade [3]. Targeted inhibitors of K-ras point mutations are under development and are either in early stages or have been complicated by emerging drug resistance [4]. The path to improved outcomes relies on targeting factors that are downstream of or cooperate with K-ras.

KM-LUAD is notable in that K-ras activation in the setting of the lung can lead to a chronically inflamed tumor microenvironment (TME) [5,6]. In normal inflammation, noxious stimuli induce various forms of cell death, and these cellular contents nourish the recruited immune cells that clear the debris and resolve the inflammatory trigger if possible [7,8]. Tumor-promoting inflammation differs from this transient response in that prolonged, chronic exposure to the noxious stimuli (e.g., cigarette smoke, infection) prevents full clearance, setting the stage for tumor initiation [8]. K-ras mutations correlate with increased myeloid lineage immune cell infiltration, including macrophages and neutrophils skewed towards pro-tumor phenotypes [5,9]. In addition to these cell types, the KM-LUAD TME fosters production of interleukin 6 (IL-6), a pleiotropic cytokine with many context-dependent effects [10,11,12,13]. IL-6 utilizes a well-known signaling pathway which results in the activation of the signal transducer and activator of transcription 3 (STAT3) [14]. STAT3 represents a critical inflammatory transcription factor in KM-LUAD, with patients with KM-LUAD tumors expressing high levels of STAT3 and activated phospho-STAT3 (p-STAT3), which correlate with worse survival [15,16]. STAT3 controls the transcription of numerous target genes that promote cell survival, angiogenesis, stemness, etc., but alarmingly, STAT3 has the ability, in concert with other transcription factors, to stimulate the production of IL-6 [13,17,18]. This feedback mechanism ensures a repeating loop of tumor-promoting inflammation. Moreover, STAT3 leads to the transcription of other soluble mediators that reprogram the TME towards a pro-tumor phenotype [19].

Given the difficulties in targeting K-ras driver mutations, tumor-promoting inflammation and the STAT3 pathway are gaining attention as therapeutic targets. However, STAT3 plays key functions across the body, as evidenced by the fact that STAT3 global deletion is early lethal [20]. Furthermore, in the context of the lung environment, STAT3 is expressed in lung resident/infiltrating immune cells and in the lung epithelium, and its functions in these cellular compartments differ yet work in synch [21]. Future therapeutics must take these cell types into account in order to produce maximal benefits. Here, we discuss the principles of STAT3 signaling and how they differ across the cell types found in KM-LUAD.

## 2. STAT3 Structure and Signaling: The Basics

The structure of STAT3 contains several functional components: a Src 2 homology (SH2) domain for recruitment and dimer formation, a C-terminal transactivation domain harboring a key phosphorylation site for STAT3 activation (Y705), a nuclear localization sequence (NLS), a DNA binding domain to recognize STAT3 inducible elements (SIEs), and a coiled-coiled domain for protein–protein interactions [22].

STAT3 is activated downstream of a plethora of receptor/non-receptor tyrosine kinases and G proteins, with ligands varying from growth factors to cytokines; however, STAT3 is classically regulated by IL-6 (Figure 1). IL-6 binds to a multimeric receptor complex comprising the IL-6 receptor (IL-6R) and gp130. Janus kinases (JAKs) 1 and 2 are recruited to the cytosolic tails of the receptors using SH2-like domains, where they transphosphorylate one another. STAT3 molecules then recognize the phosphorylated sites using their SH2 domains and are themselves phosphorylated at the Y705 site. This posttranslational modification enables STAT3 to form homodimers (or heterodimers with STAT1/5) and translocate to the nucleus, where the dimers bind SIEs to drive transcription [22,23]. In addition, IL-6R can be proteolytically cleaved from the cell surface to form soluble IL-6R (sIL-6R) [24]. sIL-6R can then induce trans signaling in cells that do not endogenously express IL-6R [24].

A host of target genes for STAT3 have been identified and have been extensively reviewed elsewhere [19], but the main responsive genes can be divided into several functional categories: anti-apoptosis, angiogenesis, invasiveness, cell cycle progression, and stemness (Figure 1). These SIE-directed programs give STAT3 signaling a decidedly pro-survival, pro-tumor flavor, which is advantageous in the setting of KM-LUAD. However, STAT3 also has the ability to modulate immune-related genes, including a mixture of pro- and anti-inflammatory mediators. These immune-related genes can lead to different immune phenotypes and as such are context and cell-type dependent; we explore the immunological specifics of these mediators in the next section. STAT3 heterodimers can also attenuate other immune signaling pathways indirectly. STAT1:STAT3 heterodimer formation is known to inhibit classical STAT1/IFN-γ signaling by “soaking up” STAT1 and preventing STAT1 homodimerization [23].

STAT3 transcriptional activity is also subject to other key post-translational modifications beyond p-Y705. S727 phosphorylation, usually mediated by ERK or other kinases, is generally believed to lead to a stronger realization of STAT3 transcriptional activity [25,26]. Acetylation, methylation, and SUMOylation of STAT3 are also regular events which have recently garnered more attention and which we will discuss in later sections [26,27]. However, it should be noted that unphosphorylated STAT3 (U-STAT3) can also translocate to the nucleus to effect changes in gene expression as a delayed response to IL-6 signaling [28].

STAT3 transcriptional targets are not limited to SIE-regulated genes. Using its coiled-coiled domain, STAT3 can interact with other transcription factors to expand its target gene repertoire. Famously, STAT3 interaction with the pro-inflammatory NF-κB transcription factor leads to expression of IL-6 [18]. This ability to induce IL-6 creates a positive feedback loop which can stimulate more and more STAT3. To counteract this loop, STAT3 also negatively regulates its own expression by transcribing the suppressor of cytokine signaling 3 (SOCS3) [29]. Moreover, phosphatases and protein inhibitors of activated STAT (PIAS) families work to nullify p-STAT3 signaling, the result of which is a relatively short half-life for p-STAT3 molecules [30,31,32].

Now that we have an overview of the mechanisms of STAT3 signaling, we explore the intricacies of this pathway in the cell types found in KM-LUAD.

## 3. Studying STAT3 by Cell Type in KM-LUAD Models

Multiple cell types comprise the KM-LUAD TME, principally tumor and immune cells. To study STAT3 signaling in these cell types in a preclinical setting, several models are employed (Table 1).

LUAD-derived cell lines, many of which display heightened STAT3 pathway activity, can represent the tumor component of the TME in vitro [33]. Coculture and/or gel matrix culture with immune cells can help to recapitulate the cell–cell interactions in the TME. These cells can be engineered to knock down or overexpress STAT3, and drugs against STAT3 or other cooperating pathways can be pioneered.

In vivo models offer a more realistic glimpse of the TME. STAT3 global knockout is early lethal, but a variety of conditional knockout systems can be used [20]. In lung tissue, Cre recombinase expression can be directed by the SPC or CCSP promoters, with SPC expressed more broadly and CCSP more tightly targeted to the conducting airway epithelium, where most KM-LUAD tumors arise [6,34]. The Cre can be temporally controlled (e.g., doxycycline-induced) if there are concerns about the timing of Cre expression [35]. Mutant K-ras is then expressed in a Cre-dependent manner and will kickstart tumorigenesis [36]. Additional genes of interest can be floxed to conditionally delete them within tumors and the lung epithelium. The same constructs can be introduced by viral vectors into the airways of adult mice, with the main side effect being potential antiviral immune artifacts [37,38]. On top of these genetic models, cigarette smoke, vaping, nicotine-derived nitrosamine ketone (NNK), and urethane can be administered to mimic smoking [39]. Products of certain bacterial species (e.g., Nontypeable *Haemophilus influenzae* (NTHi) lysate) can be introduced to recapitulate chronic obstructive pulmonary disease (COPD) and the pro-tumor inflammation which it engenders [6,39,40]. To study STAT3 in immune cells, a Cre that targets immune cell compartments can be used, most notably LysM^Cre^, a myeloid-specific driver [41].

These models have been used to a great extent to dissect the cell type-specific roles of STAT3 in KM-LUAD and are referenced in the sections that follow as we look at STAT3 functions cell by cell within the TME (Figure 2).

## 4. Cell-Specific Functions of STAT3 in KM-LUAD

### 4.1. Lung Epithelium/Tumor Cells

In healthy lung tissues, STAT3 plays important roles in lung development and differentiation, including branch morphogenesis and lung epithelial cell identity [42,43]. STAT3 is also a crucial component of the epithelial response to infection. STAT3 activation engages pro-survival machinery to reduce cell death while simultaneously triggering immune responses. The epithelium itself shows STAT3-mediated protective effects against a wide array of pathogens, but it is also able to secret immune mediators that activate features of both mucosal and suppressive immunity to preserve lung architecture and, consequently, lung function [44,45].

LUAD arises from distal regions of lung epithelium [46]. Once K-ras is mutated, cellular proliferation runs rampant as pathways such as RAF are hyperactivated [3]. At this time, STAT3 can be activated intrinsically via K-ras downstream kinase activity at the S727 site or extrinsically by cytokine signaling [26,47,48,49].

STAT3 target genes can improve the survival, proliferation, and immune evasion of early cancer cells. The Bcl and cyclin D family of genes are downstream of STAT3 and block apoptosis and drive cell cycle progression, respectively [50,51]. Moreover, the self-renewal ability of lung cancer stem cells appears to be driven in part by STAT3 [19]. STAT3 also increases the expression of immune checkpoint molecules including PD-L1, which serves to shield the cancer from antigen-specific T cell immunity [52]. Additionally, there is increasing interest in the ability of phosphorylated and acetylated STAT3 to localize to the mitochondria (Figure 1), where it increases mitochondrial gene transcription and ATP production, aiding the Warburg Effect in cancer cells [53]. Moreover, STAT3, in conjunction with NF-κB, can activate the IL-6 positive feedback loop and potentiate an autocrine signaling cycle [13]. At the same time, lung cancer cells often downregulate STAT3 inhibitor SOCS3 which exacerbates the IL-6/STAT3 loop; overexpression of SOCS3 and PIAS can restore normal tumor suppression ability [54,55]. Studies have shown that overexpression of constitutively active STAT3 in murine alveolar type II cells is sufficient to induce a strong inflammatory response and produces LUAD-like tumors [56,57]. 

Hyperactivation of STAT3 signaling also leads to the paracrine secretion of many STAT3-regulated elements and, in turn, a shift in the TME. Matrix metalloproteinases (MMPs) are able to break down the extracellular matrix to improve invasion and metastasis, and nascent tumors gain increased blood flow through VEGF-induced angiogenesis [19]. STAT3 also directly binds to the promotors of two immunosuppressive cytokines, IL-10 and TGF-β, which skew several immune cell populations from pro- to anti-inflammatory [19]. Simultaneously, pro-inflammatory IL-23 is produced and alters T cell functional phenotypes towards pro-tumor, which we explore later [19]. 

While STAT3 hyperactivation is largely believed to encourage tumor development and progression, this paradigm is clouded by some data suggesting a tumor suppressor role for STAT3. One study has found that STAT3 in the lung epithelium maintains epithelial characteristics and discourages epithelial-to-mesenchymal transition (EMT) [43]. Another group, using an adenoviral Cre model, observed a tumor suppressor function for STAT3 [58]. Still, other work suggests that STAT3 functions as a tumor suppressor in lung tumor initiation but behaves as an oncogene in established tumors [59]. Our group has also identified a sex disparity in which STAT3 deletion in the lung epithelium of a KM-LUAD mouse model is beneficial for females but harmful for males [60]. This sex disparity is characterized by increased NF-κB signaling in males lacking epithelial STAT3, which drives tumor-promoting inflammation. Introducing exogenous estrogen improved outcomes for males, indicating a potential mechanism for estrogen and estrogen receptors to mitigate NK-κB-modulated inflammation [61]. Correlations have been made with LUAD incidence by sex, with female patients experiencing a greater incidence of lung cancer but better overall survival compared to males [62]. Additionally, a recent study suggests that U-STAT3 can suppress lung cancer tumorigenesis through heterochromatin silencing of cell growth genes [63]. While these stories are still unfolding, it is necessary to avoid a purely negative view of STAT3 and pay attention to chronology, sex, or other factors that may influence the role of STAT3 in tumor development and progression.

On the whole, STAT3 hyperactivation in KM-LUAD cancer cells confers survival advantages and initiates a cascade of autocrine and paracrine signaling that alters the TME. Secreted factors in the TME interact with and recruit immune cells, where STAT3 signaling also plays crucial roles in generating tumor-promoting inflammation and immunosuppression.

### 4.2. Cancer-Associated Fibroblasts

Cancer-associated fibroblasts (CAFs) are stromal, non-malignant, non-immune lineage cells that contribute to the composition of the TME [64]. IL-6/STAT3 signaling is known to be active in CAFs, and a portion of the IL-6, VEGF, and MMPs produced in the TME is CAF-derived [65,66]. In vitro work suggests that STAT3 CAF signaling can enhance the metastatic potential of lung cancer cells, and IL-11/STAT3 signaling may play a role in cisplatin chemoresistance [66,67].

### 4.3. Macrophages

Macrophages comprise a major portion of the immune cell compartment in the lung. In homeostatic conditions, macrophages are responsible for clearing debris and pathogens to maintain gas exchange function [45]. Lung macrophages are classified based on their physical location within the lung (alveolar or interstitial) and whether or not they are classically or non-classically activated (M1 or M2 respectively) [68,69]. As the front lines of mucosal immunity, lung macrophages are programmed differently than in other tissues: to maintain lung homeostasis, lung macrophages phagocytose debris and pathogens without producing pro-inflammatory mediators, in line with an M2-like phenotype. In fact, these macrophages are major sources of the immunosuppressive cytokines IL-10 and TGF-β in the lung milieu [68].

While IL-10 and TGF-β are important in lung homeostasis, they are a decidedly pro-tumor component in the context of the TME. As previously mentioned, STAT3 signaling in tumor cells leads to the secretion of IL-10 and TGF-β, and this combined with tumor-associated macrophage (TAM) signaling creates a strongly immunosuppressive environment for existing and newly recruited immune cells. Moreover, these M2 macrophages produce VEGF, growth factors, and T cell metabolic inhibitors: arginase-1 and indoleamine 2,3-dioxygenase (IDO) [70,71].

Studies suggest that M2 polarization is in part driven by STAT3 expression. STAT1:STAT3 heterodimerization hampers the anti-tumor M1 phenotype [71]. Genetic ablation of STAT3 in macrophages results in increased pro-inflammatory cytokine production and increased antigen presentation to T cells [72]. In a urethane-induced lung cancer model, myeloid STAT3 deletion resulted in improved anti-tumor immunity, with M2 macrophages replaced by the M1 phenotype [73]. However, at least one study reports that macrophage and myeloid STAT3 hyperactivation offers anti-tumor benefits. In this study, the authors deleted the STAT3 negative regulator SOCS3, which resulted in unchecked STAT3 signaling in macrophages but overall anti-tumor effects [74]. These experiments, however, utilized a metastatic melanoma model rather than an in situ model of lung cancer. As such, the results may not translate to lung cancer development and progression in the context of tumor-promoting inflammation. In our KM-LUAD model, we see the majority of immune cells in the lung are macrophages and that they skew towards an M2 phenotype [15].

### 4.4. Neutrophils

Neutrophils are the first recruited immune cells during lung infection and are specialists in phagocytosis; however, they become a hinderance in the context of LUAD. Neutrophils are an abundant immune cell type in lung cancers, and we and others found a correlation with K-ras mutations and neutrophil recruitment in KM-LUAD models and non-small cell lung cancer (NSCLC) at large [6,75,76]. Like macrophages, neutrophils can skew between N1 and N2 phenotypes, with N2 producing TGF-β and representing a pro-tumor phenotype [77]. N2 neutrophils also parallel M2 macrophages and produce VEGF, MMPs, and reactive oxygen species (ROS) to promote tumor survival [78,79,80]. Additionally, neutrophils produce a unique immunosuppressive molecule: neutrophil elastase (NE). NE, along with heparanase and collagenase IV, remodel the extracellular matrix to promote invasion and metastasis [81]. Genetic deletion of NE in a KM-LUAD model reduced immunosuppression in the TME and reduced tumor burden [75,82].

### 4.5. Myeloid-Derived Suppressor Cells

Myeloid-derived suppressor cells (MDSCs) represent a heterogeneous group of immature myeloid immune cells which possess anti-inflammatory functions. They are further sub-characterized as either polymorphonuclear (PMN-MDSCs) or monocytic (M-MDSCs), based on their phenotypic and functional overlap with neutrophils and monocytes, respectively [83]. MDSCs are recruited by various tumor-derived chemotactic factors, including IL-8, IL-17A, and CCL3 [84]. The leading consensus is that STAT3 functions as a master transcription factor in MDSC development and function [85]. Much like M2 macrophages, MDSCs produce STAT3 transcriptional targets, including IL-10, TGF-β, VEGF, arginase-1, and IDO, to generate an immunosuppressed, pro-tumor TME [86,87]. Moreover, MDSCs induce immunosuppression with nitric oxide (NO), ROS, and PD-L1 [88,89,90]. Studies in which myeloid STAT3 is ablated show that the myeloid compartment shifts from STAT3- to STAT1-driven signaling, thereby skewing to an anti-tumor response [72].

### 4.6. T Cells

T cells are adaptive immune cells that are tailored to recognize specific targets. There are two broad classes: CD4^+^ helper T (T_h_) and CD8^+^ cytotoxic T lymphocytes (CTLs). While CTLs perform the actual work of killing tumors cell by cell, T_h_ cells produce specialized cytokines that support different flavors of immune responses: T_h_1 (antiviral/anti-tumor), T_h_2 (anti-parasite/allergy), T_h_17 (mucosal immunity), Treg (immunosuppression). Favorable cell types for targeting tumors are CTLs and T_h_1, which elicit high STAT1 signaling. These cells can directly induce tumor apoptosis via granzyme B (CTLs only) and production of IFNγ and TNFα, two cytokines which broadly reprogram the TME and can induce apoptosis in tumor cells [91,92]. On the other hand, T_h_2, T_h_17, and Tregs are generally pro-tumor. Within these diverse subtypes, STAT3 signaling is a crucial factor that skews T cells towards pro-tumor T_h_17 and Treg immunity while also increasing PD-L1 expression, yielding functional exhaustion [52,93].

T_h_17 differentiation is triggered in response to IL-6 and TGF-β, two cytokines which are downstream of STAT3 and which are found in abundance in the KM-LUAD TME [93]. These cytokines lead to activation of STAT3, which activates the ROR family of transcription factors to drive T_h_17 fate [94]. T cells that lack STAT3 cannot become T_h_17 cells and instead differentiate into T_h_1 cells [95]. Fully fledged T_h_17 cells are later maintained by IL-23, another STAT3-controlled cytokine found in the TME [96].

True to their name, T_h_17 cells produce IL-17A, a cytokine which leads to production of IL-8 [97]. Both IL-17A and IL-8 are major chemotactic factors for neutrophils and MDSCs [84,98,99]. Moreover, IL-17A signaling contributes to paracrine TME reprogramming, increased inflammation, and angiogenesis [100,101]. In our own studies, deletion of IL-17A in a KM-LUAD mouse model reduced tumor burden, inflammation, and pro-tumor myeloid cell recruitment and infiltration [102]. T_h_17 cells can also inhibit other anti-tumor T cells by cleaving extracellular ATP into adenosine, which is immunosuppressive [103].

It is important to note that non-classical γδ T cells and type 3 innate lymphoid cells (ILC3s) are major producers of IL-17A [104]. However, ILC3s have been reported to form tertiary lymphoid structures (TLSs) in lung cancer that are positively prognostic [105]. It should be noted that IL-17A can also play an anti-tumor role, depending on the type of cancer [106]. In additional to making IL-17A, CD4^+^ and γδ T cells also produce IL-22, a cytokine which acts on non-immune cells and heavily activates the STAT3 pathway [107]. Lung cancer patients present with elevated IL-22 in lung lavage samples [108]. We have shown that deletion of IL-22 in a KM-LUAD mouse model suppressed tumorigenesis, altered the TME to an anti-tumor phenotype, and reduced stemness properties of tumor cells [109]. 

Treg differentiation is driven by exposure to TGF-β and subsequent activation of FoxP3, a transcription factor that induces and maintains the Treg subtype [110]. Tregs can express immune checkpoint molecules that directly inactivate neighboring effector T cells. They can release immunosuppressive molecules including IL-10, TGF-β, adenosine, etc. They also metabolically compete with anti-tumor immune cells [111]. Treg infiltration is almost universally a negative prognostic marker across cancer types [112]. When knocking down STAT3 in T cells, Tregs are less abundant, indicating a reliance on STAT3 [113].

### 4.7. Dendritic Cells

Dendritic cells (DCs) are a key bridge between innate and adaptive immunity and prime antigen-specific T cells [114]. The various DC subtypes and their roles in cancer and immunotherapy have been extensively reviewed elsewhere [115], but the principal DC subtypes of relevance are classical DC type 1 (cDC1) and 2 (cDC2). These cDC types largely correlate with T_h_ subsets: cDC1s are considered anti-tumor and prime tumor-specific CTL and T_h_1 responses. cDC2s, on the other hand, are more heterogeneous in function, but they are largely thought to prime T_h_2, T_h_17, and Treg responses, which represents a pro-tumor skewing.

DC maturation often depends on STAT3 [116,117]. However, deletion of STAT3 in the DC compartment leads to fewer but more functional DCs capable of activating robust T cell responses [41]. STAT3-ablated DCs demonstrate increased IL-12 secretion necessary for T_h_1 differentiation, and tumors with STAT3 inhibited DCs show decreased Treg infiltration with commensurate increases in intratumoral CTLs and repolarization toward type I immunity [41,118]. Interestingly, acetylated STAT3 has been implicated as a precursor to IDO production in DCs, suggesting an additional immunosuppressive mechanism for STAT3 [119]. 

### 4.8. Natural Killer Cells

Natural killer (NK) cells are directly cytotoxic immune cells that recognize their targets independent of MHC presentation [120]. NK function is largely inhibited in KM-LUAD by immunosuppressive factors such as IL-10, TGF-β, and IDO [121]. Deletion of STAT3 in the NK compartment drastically improved tumor rejection, immune surveillance, and IFN-γ secretion abilities, and targeting STAT3 with microRNA-130a augmented NK cytotoxicity against NSCLC cells [122,123]. Similarly, deletion of lung epithelial STAT3 in a urethane-induced LUAD model improved NK cell tumor killing function [57].

### 4.9. B Cells

B cells have recently come under investigation in lung cancer, displaying both pro- and anti-tumor traits. STAT3 appears to be necessary for regulatory B (Breg) cells, a B cell counterpart to Tregs that produces anti-inflammatory IL-10 and is pro-tumor [124]. B cell-specific deletion of STAT3 causes severe autoimmunity in mice, so we can infer potential anti-tumor benefits from targeting STAT3 signaling in B cells [125]. Moreover, Bregs are found more frequently in lung cancer patients [126]. One study suggests that STAT3-regulated CD5 expression on B cells drives IL-6 signaling and disease progression in lung cancer [127].

On the other hand, B cells have the ability to form structures which histologically and functionally resemble lymph nodes and lymphoid tissue, so-called tertiary lymphoid structures (TLSs). TLSs are associated with increased Th1 polarization, CTL infiltration, and DCs in NSCLC. Much like true lymphoid tissue, TLSs develop germinal centers, a process which depends on STAT3 [128].

## 5. STAT3 Inhibition

In light of the many negative effects of hyperactive STAT3 in the TME, a number of therapeutic interventions have been developed that target different stages of STAT3 signaling. Many of these drugs are already in or finishing clinical trials and have been extensively reviewed elsewhere [129,130]. However, we want to touch on a few of these which hold the most promise.

Multiple small molecule STAT3 inhibitors have been tested in the last decade, and there are still many novel inhibitors which may one day reach clinical application [130,131]. One such STAT3 inhibitor, TTI-101, binds to the SH2 domain to prevent STAT3 homodimerization [132]. TTI-101 has shown preclinical benefits in various mouse cancer models including lung cancer xenografts, and recruitment for clinical trials has begun (NCT03195699) [133]. AZD9150 is an anti-sense oligonucleotide therapy targeting STAT3 on the RNA level. AZD9150 improves anti-tumor immunity in patient-derived lung cancer xenografts [134]. Multiple clinical trials are underway to combine AZD9150 with an immune checkpoint blockade (NCT02983578, NCT03334617, NCT03421353).

It is easier to block cytokines than to target a transcription factor. Guided by this philosophy, a monoclonal antibody (mAb) against IL-6, ALD518, completed phase II trials in 2011 in patients with advanced NSCLC (NCT00866970). ALD518 helped patients maintain body weight while reducing fatigue and cancer-related anemia [135]. A forthcoming clinical trial will test the synergistic effects of Tocilizumab (anti-IL-6) and Atezolizumab (anti-PD-L1) in patients with NSCLC who have previously failed immune checkpoint blockade and are experiencing advanced or metastatic disease (NCT04691817).

## 6. Conclusions

Despite advances in cancer therapies, there is rarely a one-treatment solution, and KM-LUAD is no exception. To study the complex cell types, signaling pathways, and possible treatment combinations, we need to make full use of the available models that exist and to test treatments that attack multiple aspects of the TME. KM-LUAD models are vital to understanding the mechanisms of STAT3 and to appraise therapies that block the IL-6/STAT3 signaling pathway across the variety of cancer and immune cells within the TME. The importance of cell types will feature prominently in future treatment strategies. For instance, global STAT3 targeting in the TME may be beneficial in the T cell compartment but harmful for B cells; ways to deliver treatments in a targeted manner will take advantage of the diversity within the TME to yield optimal anti-tumor responses. In our own group, we have also seen that targeting STAT3 in tumors depends greatly on the biological context, with sex hormones playing a greater role than we foresaw. By cell type-specific interrogation of the TME, we are creating a mosaic map and learning how to better tailor future interventions.

## Figures and Tables

**Figure 1 cancers-14-01785-f001:**
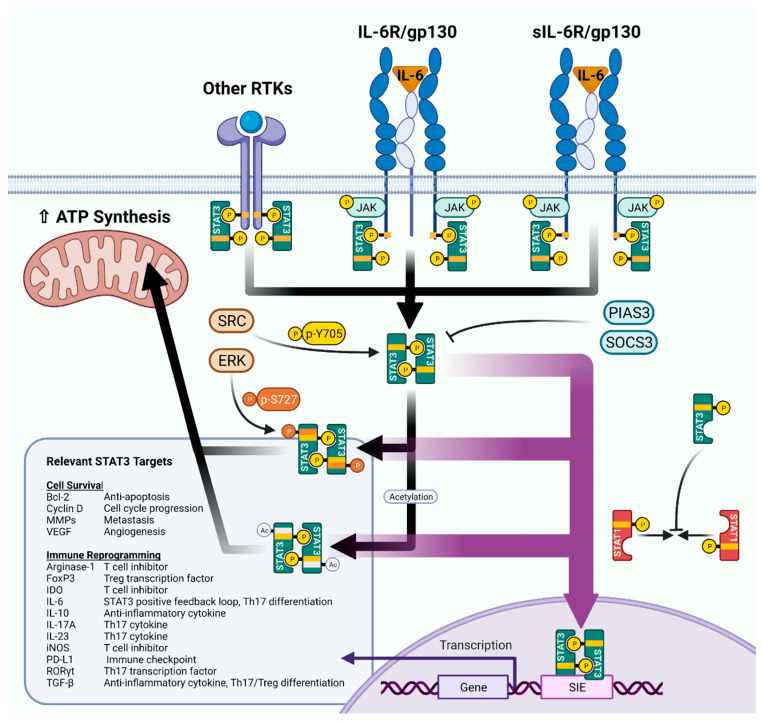
An overview of STAT3 signaling. Figure created in BioRender.

**Figure 2 cancers-14-01785-f002:**
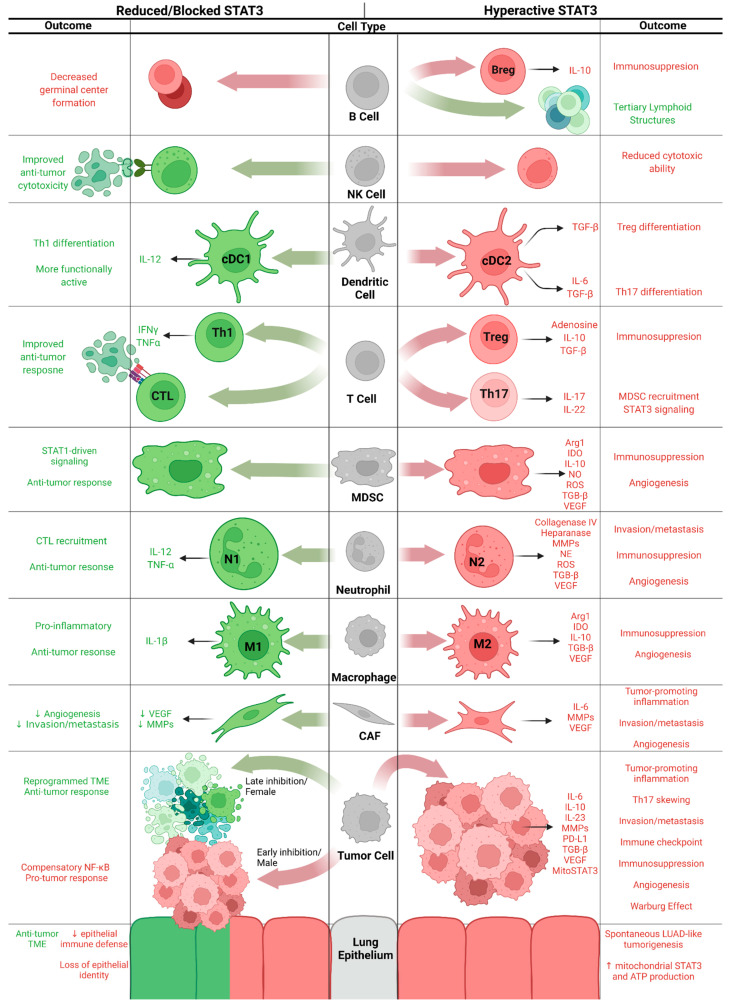
Cell type-specific effects of hyperactive and attenuated STAT3 signaling, with anti-tumor effects in green and pro-tumor effects in red. Figure created in BioRender.

**Table 1 cancers-14-01785-t001:** Mouse models available to study STAT3 in KM-LUAD.

Approaches	Examples	Induction Method
Tissue-Specific Cre Recombinase	CCSP^Cre^: lung epithelium	
	SPC^Cre^: whole lung rtTA-tetO: dox-inducible	Germline/AdCre
	LysM^Cre^: myeloid CD4^Cre^: helper T cell CD19^Cre^: B cell NCR1^Cre^: NK cell	Germline/Adoptive Transfer
Mutations	Kras^LA2/+^ Kras^LSL-G12D/+^ Kras^LSL-G12V/+^ ±p53^lox/lox^	Germline/AdCre
COPD-like Inflammation	NTHi LPS	Nebulization Intranasal
	Cigarette Smoke Elastase	Smoke exposure Intratracheal
Carcinogen	NNK Urethane	Intraperitoneal
	Cigarette Smoke eCigarette	Smoke exposure Atomized nicotine
Xenograft	Immunocompromised mouse	Orthotopic, subcutaneous, or intravenous

AdCre: Adenoviral delivery of Cre and any other transgenes.

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
