# Peer review of "Cell Type-Specific Roles of STAT3 Signaling in the Pathogenesis and Progression of K-ras Mutant Lung Adenocarcinoma"

_cancers, 2022, doi:10.3390/cancers14071785_

Round 1
Reviewer 1 Report
This study investigates the role of signal transducer and activator of transcription 3 (STAT3) in Kras Mutant Lung Adenocarcinoma (KM-LUAD) progression, and suggests that this could serve as a new target for preventive and therapeutic intervention for KM-LUAD. The study reported in this manuscript has significant importance to lung cancer study and should be published if the authors are willing to carry out the following minor revisions:
- Page 5, in Figure 1, the authors mentioned the pro-inflammatory cytokine TNF-alpha. The authors may discuss its role as tumor suppressor in TME.
- Page 6, line 168, “serve” should be “serves”
- In the manuscript, the authors mentioned "tumor-promoting inflammation" several times, which needs further elaboration. Basically, the authors need to explain what kinds of inflammations are carcinogenic.
Inflammation is the central machinery of our immune system in response to acute tissue damage [1,2], and local transient inflammation is protective. It helps to remove the injurious stimuli like infections and trauma, and initiate tissue regeneration.
When there are tissue-damaging stimuli like physical injury, infectious pathogens, toxin exposure, chemical irritation, the human immune system will actively induce cell self-destruction (programmed cell deaths like apoptosis [3,4], necroptosis [5] and pyroptosis [6]) and reuse the nutrition from the degradation of the dead cell as nutrition source (immuno-nutrition) to repair/regenerate the tissue cells. If the damaging stimuli can be removed by inflammation, then inflammation will not be carcinogenetic. Otherwise, the prolonged inflammatory response will turn out to be chronic, and it becomes the initial step of carcinogenesis [3-6].
Reference:
1. Greten FR, Grivennikov SI (2019) Inflammation and Cancer: Triggers, Mechanisms, and Consequences. Immunity 51(1):27-41. DOI: 10.1016/j.immuni.2019.06.025 2. Yang Y, Jiang G, Zhang P, Fan J. (2015) Programmed cell death and its role in inflammation. Mil Med Res. 2015;2:12. DOI: 10.1186/s40779-015-0039-0Author Response
Please see the attachment.

Reviewer 2 Report
Clowers and Moghaddam had described in this Review the cell type-specific roles of STAT3 in contribution towards Kras Mutant Lung Adenocarcinoma. Given the complexity in tackling this class of lung cancer, there is an urgent need to look into the tumor microenvironment and inflammation that promotes tumor growth. STAT3 is an attractive potential target given that it is druggable and has been relatively well established for immuno signaling. While the review is rather comprehensive covering on how STAT3 in different immune cells may contribute towards tumor growth, the Review can be potentially strengthened on the therapeutics and study methods regarding the cell-type specificity. This is especially so when the authors mentioned "Future therapeutics must take these cell types into account in order to produce maximal benefits." (Line 62/63). Here are some comments:-
- In the section "STAT3 Structure and Signaling", the authors have shown how JAK1/2 recruitment to the receptors lead to STAT3 activation in a detailed manner, and also the downstream regulation, and how other signaling pathways may interact and shape its signaling. The authors could perhaps strengthen this section with a figure depicting these points in a clearer manner. In addition, the temporal dimension of STAT3 activation and spatial dimension of STAT3 activity (plus the noncanonical activity in mitochondria when S727 is phosphorylated) should be discussed with greater detail.
- In figure 1, the authors have shown clearly the effects of STAT3 blockage or hyperactivation, and how these events in the different cell types lead to apoptosis or survival/stemness. Further description on how the different cell types contribute to tumor would help readers visualize the multicellular interaction in the process, especially on the temporal sequence and positive feedback loop.
- STAT3's controversial role as a potential tumor suppressor should also be depicted in Figure 1 to avoid oversimplifying the picture.
- Discussion on STAT3 inhibition should be further expanded. Details about the therapies and their efficacies and clinical progress should be summarized in a table. The knowledge of STAT3 activity in each cell type can also be discussed as a value-add for potential new therapeutic treatment and also implications when used in combinatorial treatment.
